# Greengage Grading Method Based on Dynamic Feature and Ensemble Networks

**Keqiong Chen** [1,*] , **Weitao Li** [2] , **Jiaxi An** [1] and **Tianrui Bu** [1]

1   School of Advanced Manufacturing Engineering, Hefei University, Hefei 230601, China;
    anjx@hfuu.edu.cn (J.A.); butr@hfuu.edu.cn (T.B.)
2   School of Electrical Engineering and Automation, Hefei University of Technology, Hefei 230009, China;
    wtli@hfut.edu.cn
*   Correspondence: chenkq@hfuu.edu.cn; Tel.: +86-137-3922-0668

**Abstract:** To overcome the deficiencies of the traditional open-loop cognition method, which lacks evaluation of the cognitive results, a novel cognitive method for greengage grading based on dynamic feature and ensemble networks is explored in this paper. First, a greengage grading architecture with an adaptive feedback mechanism based on error adjustment is constructed to imitate the human cognitive mechanism. Secondly, a dynamic representation model for convolutional feature space construction of a greengage image is established based on the entropy constraint indicators, and the bagging classification network for greengage grading is built based on stochastic configuration networks (SCNs) to realize a hierarchical representation of the greengage features and enhance the generalization of the classifier. Thirdly, an entropy-based error model of the cognitive results for greengage grading is constructed to describe the optimal cognitive problem from an information perspective, and then the criteria and mechanism for feature level and feature efficiency regulation are given out within the constraint of cognitive error entropy. Finally, numerous experiments are performed on the collected greengage images. The experimental results demonstrate the effectiveness and superiority of our method, especially for the classification of similar samples, compared with the existing open-loop algorithms.

**Keywords:** greengage grading; dynamic feature; entropic constraint; feedback regulation; deep ensemble learning

## 1. Introduction

The realization of the automatic classification of fruit grades has become an essential precondition for the modernization of the fruit industry [1]. Greengage is a kind of pharmaceutical and food resource with multiple healthcare functions which is favored by the masses. At present, the existing automatic classification mostly involves screening of the particle size and weight. The sorting of its quality often relies on manual screening, which is not only labor-intensive but also susceptible to subjective factors such as operator experience, so its cognitive effect is hard to evaluate satisfactorily. Therefore, development of a fast and accurate machine grading method becomes an urgent need to promote the fruit industry [2–4]. A fast classification method for fruit grading based on multiple kernel support vector machines (kSVM) is proposed in [2]. A fuzzy cluster-based image segmentation method is proposed in [3], and the extracted features are introduced into the deep neural network to achieve apple grading. In [4], a carrot surface defect detection method based on the fusion of computer vision and deep learning was proposed to achieve real-time carrot quality grading. Various levels of the feature space and various perspectives within the same feature level represent discriminative attention. However, with uncertain image inputs and indeterminate grade outputs, the traditional machine fruit grading methods with the open-loop method lack updated data structures of the feature space and classified

criteria once established, which is an obvious difference from the human information interaction mechanism with repeated comparison and inference.

Deep learning can build neural networks that imitate analysis and learning from the global to local levels with the human brain. As a typical method, the CNN has received widespread attention [5–8]. However, the traditional cognitive methods with CNNs still belong to the open-loop mode. Generally, increasing the network level can reduce the feature space dimension and extract more detailed information, while the expressive ability of the features is proportional to the number of feature maps in the same network level within a certain range. Nevertheless, too many network levels and feature maps will result in an enormous increase in redundant features and computational complexity. Therefore, the joint training mechanism between deep feature extraction and a classification model has become a popular research direction [9]. Furthermore, the random initialization of the weight and the uncertainty of the structure would lead to an unstable output during network modeling. Thus, ensemble network training can overcome the random deviation of the model to a certain extent and improve the generalization performance of the model.

The human cognition mechanism is a hierarchical information processing process of repeated comparison and inference with prior knowledge. That is to say, a system with a feedback mechanism could make optimal decisions by performance evaluation, which imitates the human cognition characteristics in a sense. However, the performance of the traditional feedback system is usually measured by the index functions. It is difficult to define a unified performance index for an intelligent cognition system. Entropy is usually used to establish the index model and optimize the system performance [10–12]. An inherent fuzzy entropy-based algorithm is proposed in [10] to achieve a more reliable electroencephalogram (EEG) complexity assessment. A multi-label maximum entropy (MME) model is introduced to realize emotion classification over short text in [11]. An improved SVD entropy-based feature reduction method is proposed in favor of the related feature selection in [12]. Therefore, as with the employment of the intelligent control theory of Saridis [13], the index model with the form of an entropy function can be used to evaluate the cognitive performance so as to construct a humanoid feedback mechanism. In this way, the entropy function is adopted as the unified performance index for hierarchical greengage grading to establish the equivalent measurement relationship between the information theory and optimal cognition problem.

In [9], a method for greengage grading is proposed to overcome the deficiencies of a traditional open-loop system, which lacks evaluation of the uncertain inputs and outputs. Nevertheless, the joint feedback mechanism, either semi-supervised or supervised, can learn more cognitive knowledge with fewer labeled samples while increasing the complexity of the learning model. In practical agricultural production applications, the learning and decision-making efficiency is one of the most important system performance indicators. In addition, feature selection based on the Mahalanobis distance has a weakness: performance in nonlinear space processing. From the perspective of information, more important cognition knowledge can be picked out to enhance the discriminative power of the compact feature space. At the same time, hierarchical labeled confidence thresholds are set for the semi-supervised mechanism in [9] to realize feedback adjustment. However, such a feedback mechanism is still quite different from the human cognition mechanism. In fact, a hierarchical feature space can be constructed by considering different granular levels of information in feature selection, and error feedback adjustment would be realized to obtain a distributed cognitive knowledge space so as to effectively improve the system performance.

Therefore, to expand our previous work, the technical contributions in this paper are summarized as follows: (1) proposing a greengage grading model with dynamic feature and ensemble networks to improve the accuracy of the existing methods, (2) introducing an SCN-based bagging network model to enhance the robustness of the greengage grade classifier, and (3) constructing the feedback regulation criteria based on the classified accuracy and CNN level to imitate the human cognition mechanism with repeated comparison

and inference from macro to micro. First, a cognitive architecture for greengage grading with an adaptive error feedback mechanism is established to imitate the human cognition mechanism, and the functions of each layer are analyzed. Secondly, an optimal dynamic expression model of a convolutional feature is established, and then an ensemble deep SCN is constructed to realize the optimal representation of greengage images with sufficiency and separability. Third, the cognitive error of the greengage grade is represented with the form of entropy, and then the regulation criteria of the feature level and feature efficiency are established based on the constraint of cognitive error entropy. Ultimately, the intelligent greengage grading algorithm based on dynamic feature and ensemble networks is given to imitate the intelligent human cognition mechanism with repeated comparison and inference from macro to micro. Finally, numerous experiments with comprehensive comparisons are carried out based on random greengage images. The experimental results demonstrate the effectiveness and superiority of our method compared with existing open-loop algorithms.

## 2. Framework of the Cognitive Model for Greengage Grading Based on Dynamic Feature and Ensemble Networks

Targeting the problem that a significant difference exists in the traditional open-loop model for automatic greengage grading and the human information processing mechanism with repeated comparison and inference, a novel cognitive model for greengage grading based on dynamic feature and ensemble networks is proposed in this paper. With our cognitive model, the cognitive knowledge for greengage grading is dynamically and hierarchically represented with sufficiency and separability to realize the repeated cognition of greengage samples in a finite domain. The framework of the proposed model is shown in Figure 1. A three-layered interconnected structure, comprising a training layer, cognitive layer, and feedback layer, is adopted to achieve real-time information interaction between training and cognition.

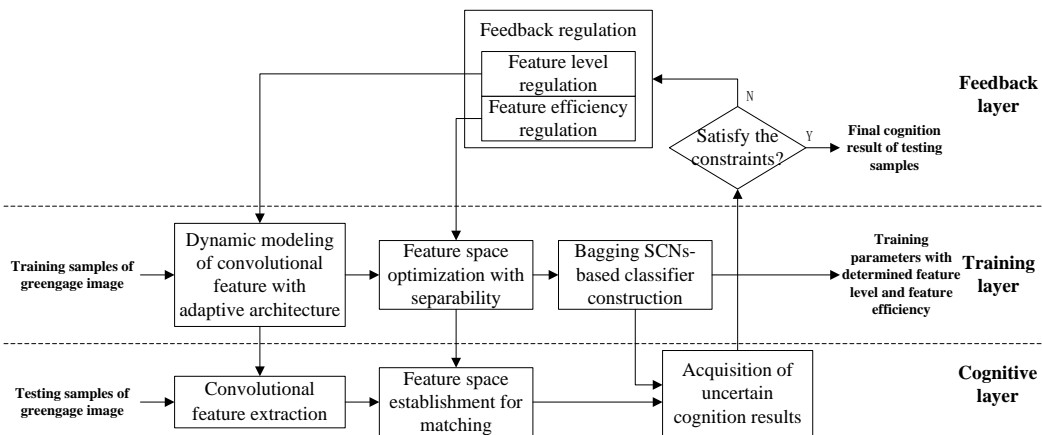

**Figure 1.** The cognitive model for greengage grading based on dynamic feature and ensemble networks.

1　　Training Layer

Based on the parameters of the feature level and maximum entropy given by the feedback layer, the convolutional feature space of the training greengage images with sufficient information is dynamically modeled. Based on the parameter of feature efficiency issued by the feedback layer, an information system for greengage grading with separable features with a determined feature level and feature efficiency is established. Based on the bagging algorithm, the ensemble SCN classifier is constructed to obtain the learning results of the training space for greengage grading, and then the integrated errors are backpropagated to optimize the model parameters for greengage grading with a determined feature level and feature efficiency overall. The classification rules are ultimately established for greengage grading within the cognitive layer.

2. Cognitive Layer

According to the feature modeling and classification rule construction provided by the training layer, the optimal feature space of the testing greengage images is dynamically established so as to obtain uncertain cognition results for greengage grading with a determined feature level and feature efficiency. The heuristic comparison knowledge within the feedback layer is provided to evaluate the uncertain cognition process and the results for greengage grading.

3. Feedback Layer

Based on the knowledge information provided by the cognitive layer and to imitate human cognition mechanism, the indexes in the form of an entropy function are constructed to measure the credibility of the cognition results of the test database. The feedback cognition mechanism for greengage grading is constructed within the entropic constraint of the cognitive error, and then the network level and the classified accuracy, which represent the feature level of the interlevel and feature efficiency within the hierarchy, respectively, are self-optimizing and regulated to achieve fast and accurate cognition macroscopically for the samples near the clustering center and the hierarchical feedback cognition microscopically for the samples near the classification surface with a finite domain.

## 3. Dynamic Feature Modeling of a Greengage Image with a Deep Neural Network

### 3.1. Deep Feature Extraction of a Greengage Image with Adaptive Dynamic Optimization

Within CNNs, hierarchical feature space can be established by regulating the network depth. However, the complexity of parameter training increases with the increase in network depth. Therefore, a CNN-based dynamic feature extraction model of greengage images with adaptive structures is proposed to realize the hierarchical representation of the cognitive knowledge for greengage grading in the condition of a finite domain.

The cognitive information for greengage grading can be represented in various depths with an alternating structure of convolution and pooling. However, the traditional open-loop cognitive methods of CNNs are quite different from the human information interaction mechanism with repeated comparison and inference, and the increase in network depth will greatly increase the computational complexity during feature extraction. To imitate the human cognition mechanism, a dynamic convolution feature extraction model of a greengage image with an adaptive structure is proposed. The learning process of the training samples and the cognitive process of the testing samples are correlated to adaptively regulate the network depth of CNNs with finite domains.

Considering the consistent number of feature maps of contiguous convolutional and pooling layers, here, we treat the connected structure with a convolutional layer and a pooling layer as a feature level to build the dynamic CNN feature space. Suppose the current feature level in the $w$th feedback cognition is $l_w$. Namely, the depth of the CNN is $l_w$, consisting of $l_w$ combinations of convolution and pooling within the model. The model structure is shown in Figure 2. As we can see from Figure 2, for an input training sample of greengage image $U_i$, a feature map $m_{l_w}$ with a size $u_{P_{l_w}} \times v_{P_{l_w}}$ can be gained successively after $l_w$ alternating processes of convolution and pooling.

Here, we mainly discuss the influence of $l_w$ on the cognitive results for greengage grading with a finite domain. Therefore, the other parameters of the proposed model of convolutional feature extraction in this paper were selected as follows. The convolution kernel size is $5 \times 5$, the pooling range size is $2 \times 2$, the max pooling method is adopted with pooling operation, and the sigmoid function is used as the activation function. At this point, every additional feature level combination of CNNs will reduce the size of the feature map. Therefore, according to the original size $u \times v$ of the input image samples, the feature level $l_w$ of the CNNs should have an upper limit $l_{\max}$ in the feedback cognition process, and $l_{\max}$ should satisfy Equation (1):

$$l_{\max} < \frac{\min\{u, v\} + 4}{8} \tag{1}$$

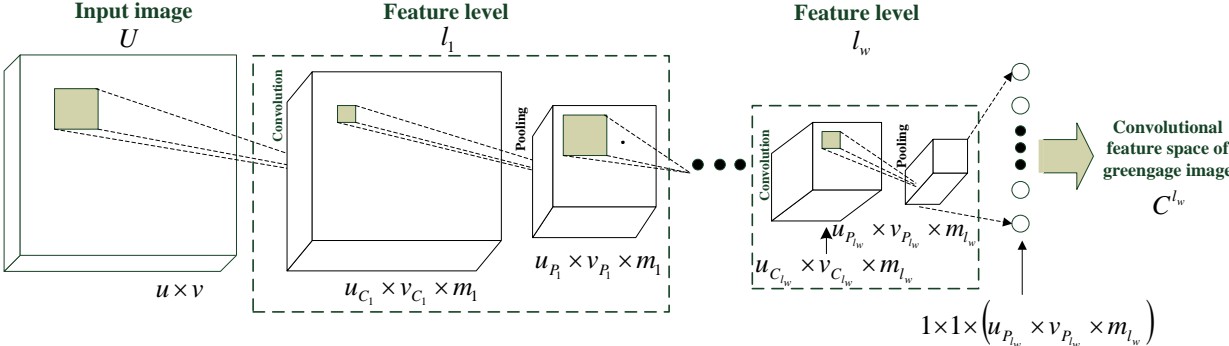

**Figure 2.** Structure of the convolutional feature extraction of a greengage image.

In addition, based on a convolutional kernel with a size of $1 \times 1$, a convolution operation is carried in the full connection layer $F_C$. By these means, the $K$ dimensional feature vector $C_i^{l_w} = \left\{ C_{i,1}^{l_w}, \ldots, C_{i,K}^{l_w} \right\}$ of $U_i$ is obtained and sent to the following cognitive section, where $K = u_{P_{l_w}} \times v_{P_{l_w}} \times m_{l_w}$ is the dimension of the extracted feature vector.

Furthermore, during convolution operation within a determined feature level, the mapping from an input to an output can be realized with a convolutional kernel. In general, more feature maps will be obtained by increasing the convolutional kernels to represent the image information from various perspectives. However, under the condition of a finite domain, increasing the feature dimensions will result in saturated information content and an enormous increase in computational complexity. Therefore, it is necessary to seek a balance between the size of the feature space and the amount of computation.

Actually, in the determined feature level $l_w$, there is a domain of a greengage feature map space $x^{l_w-1} = \left\{ x_1^{l_w-1}, \ldots, x_{m_{l_w-1}}^{l_w-1} \right\}$. According to [14], when the amount of the information content within the domain $x^{l_w-1}$ reaches its maximum, there is the largest value of information entropy $H_{\max}^{l_w-1} = \log_2 m_{l_w-1}$. In this respect, Equation (2) is constructed as the iterative constraint in feature extraction of the greengage image within the determined feature level of the CNNs so that the minimum feature map space of the greengage images $x^{l_w} = \left[ x_1^{l_w}, \ldots, x_{m_{l_w-1}}^{l_w} \right]^{\mathrm{T}}$ can be established with sufficient information:

$$
\begin{cases}
\underset{m_{l_w}}{\max} H\big( R(x^{l_w}) \big) \\
H\big( R(x^{l_w}) \big) = -\sum\limits_{q=1}^{\theta_{l_w}} \dfrac{|Z_q|}{|x^{l_w}|} \log_2 \dfrac{|Z_q|}{|x^{l_w}|} \\
s.t.\ 0 < m_{l_w} \le m_{\max}, \theta_{l_w} \le m_{l_w-1}
\end{cases}
\tag{2}
$$

where $x_i^{l_w} = \left[ x_{i,1}^{l_w}, \ldots, x_{i,m_{l_w}}^{l_w} \right]$ is the feature map vector extracted from the $i$th input of $x^{l_w}$, $i \in [1, m_{l_w-1}]$, $x_{i,j}^{l_w}$ is the $j$th feature map of $x_i^{l_w}$, $j \in [1, m_{l_w}]$, $H\big( R\big(x^{l_w}\big) \big)$ is the information entropy calculated by [13], and $R\big(x^{l_w}\big) = \left\{ Z_1, \ldots, Z_{\theta_{l_w}} \right\}$ is the quotient set divided from $x^{l_w}$ based on the equivalent relationship.

From Equation (2), the minimum feature map space with the largest amount of information in the current feature level can be established so as to realize the sufficient representation of greengage images based on the maximum information entropy. In this way, with the increase in depth of the CNNs (namely an increasing feature level $l_w$), deeper image features can be extracted to achieve dynamic representation of cognitive knowledge for greengage grading.

Because of factors such as light and the acquisition equipment, the redundant and related information must exist in the extracted feature space of a greengage image which is sufficiently established based on the maximum entropy. The traditional cognition method, which gains the output directly from the full connection layer, may lead to the curse of dimensionality. A rough set enables data mining to analyze the information of imprecise,

inconsistent, and incomplete information [15]. With the adoption of a variable precision rough set and the conditional entropy-based feature selection algorithm [16], the parameter of classified accuracy is introduced in this paper. The decision information system for greengage grading $S^{l_w,\beta_w} = \left\{ U, \boldsymbol{B}^{l_w,\beta_w} \cup \boldsymbol{D} \right\}$ with separable representation is constructed with a determined feature level and feature efficiency, where, $U = \{U_1, \ldots, U_n\}$ is the training dataset of the greengage images, $B^{l_w,\beta_w} = \left\{ \boldsymbol{B}_1^{l_w,\beta_w}, \ldots, \boldsymbol{B}_\delta^{l_w,\beta_w} \right\}$ is the simple feature space of the training samples with $\delta$ dimensions selected from $C^{l_w} = \left\{ \boldsymbol{B}_1^{l_w}, \ldots, \boldsymbol{C}_K^{l_w} \right\}$, $C^{l_w}$ is the sufficient feature space of the training samples with $K$ dimensions extracted based on $l_w$ levels of the network of CNNs, $B^{l_w,\beta_w} \subseteq C^{l_w}$ and $\delta \leq K$, $\beta_w$ are the current given classified accuracy, and $\boldsymbol{D} = [D_1, \ldots, D_o]^{\mathrm{T}}$ is the grade label of the training samples. From [16], it can be seen that for $C^{l_w}$ with a given $\beta_w$, the classified quality could map with one-to-one feature efficiency during feature space separable optimization of a greengage image so as to improve the model generalization.

### 3.2. Cognitive Rule Construction for Greengage Grading Based on Bagging Ensemble SCNs

In recent years, SCNs have been widely applied due to their advantages in learning efficiency and approximation performance [17–20]. However, the randomness of a single model would make for an unstable network output. It has been proven that ensemble modeling can improve the generalization, effectiveness, and robustness of the model [21]. The bagging ensemble method can effectively reduce the model variance while ensuring model deviation to further improve the generalization performance of the model [22]. Therefore, a bagging SCN-based classifier is constructed to build an accurate and stable cognitive criterion for greengage grading.

Considering the compactness of the model, $n_h$ sample feature subspaces are constructed with random sampling from the compact feature space of a greengage image $B^{l_w,\beta_w}$, which is gained with adaptive dynamic optimization, and then the classifier training can be realized based on the above subspace. On the basis of the classic SCN [21], the structure of a bagging SCN-based classifier for greengage grading is shown in Figure 3.

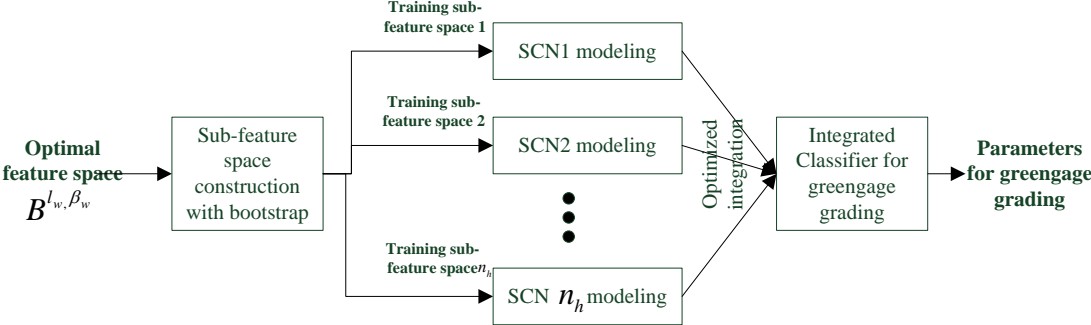

**Figure 3.** Structure of bagging SCN-based classifier for greengage grading.

For the subspace $\boldsymbol{B}_h^{l_w,\beta_w}$ ($h \in [1, n_h]$), suppose that there is an SCN with $L-1$ hidden layer nodes. Its input-output relationship is

$$f_{L-1}^h\left(\boldsymbol{B}_h^{l_w,\beta_w}\right) = \sum_{j=1}^{L-1} \mu_{h,j}\gamma_{h,j}\left(\omega_{h,j}^{\mathrm{T}} \cdot \boldsymbol{B}_h^{l_w,\beta_w} + \rho_{h,j}\right) (L = 1, 2, \ldots; f_0 = 0) \tag{3}$$

where $\boldsymbol{\mu}_h = [\mu_{h,1}, \ldots, \mu_{h,L-1}]^{\mathrm{T}}$ is the output weight, $\omega_{h,j}$ and $\rho_{h,j}$ are the input weights and biases, respectively, and $\gamma_{h,j}(\cdot)$ is the activation function with which the sigmoid function is chosen. In the SCN, $\omega$ and $\rho$ can be found from $[-\lambda, \lambda]$, and $\lambda$ is adjustable.

The output error of the current network can be expressed as

$$\begin{cases} e_{L-1}^h\left(\boldsymbol{B}_h^{l_w,\beta_w}\right) = f^h\left(\boldsymbol{B}_h^{l_w,\beta_w}\right) - f_{L-1}^h\left(\boldsymbol{B}_h^{l_w,\beta_w}\right) = \left[e_{L-1,1}^h\left(\boldsymbol{B}_h^{l_w,\beta_w}\right), \ldots, e_{L-1,d}^h\left(\boldsymbol{B}_h^{l_w,\beta_w}\right)\right] \\ e_{L-1,t}^h\left(\boldsymbol{B}_h^{l_w,\beta_w}\right) = \left[e_{L-1,t}^h\left(\boldsymbol{B}_1^{l_w,\beta_w}\right), \ldots, e_{L-1,t}^h\left(\boldsymbol{B}_\delta^{l_w,\beta_w}\right)\right] \end{cases} \tag{4}$$

For the current $B^{l_w,\beta_w}$, when the calculated network output residual $\|e_{L-1}^h\left(B_h^{l_w,\beta_w}\right)\|$ cannot meet the error requirements of the system, the hidden layer node parameters $\omega_{h,j}$ and $\rho_{h,j}$ are optimized according to the supervision mechanism until the preset conditions are met. Thus, the optimal parameters of the $n_h$ basis classifiers can be obtained. More details on the SCN can be found in [17].

The output residuals of the classifier need to be backpropagated through the feature space to update the relative parameters of the CNNs and bagging SCN in a global sense. During the backpropagation process of the output error of the ensemble classifier, assuming the cognition result of the $h$th SCN output is $X_h$. Then, the training error of the network can be expressed as $e^h = \sqrt{(X_h - D)^{\mathrm{T}}(X_h - D)}, (h \in [1, n_h])$. Therefore, the backpropagated error of the input-output is

$$ex_h = \omega_h((\hat{\mu}_h e_h^\dagger)^\dagger((\hat{\mu}_h e_h^\dagger)^\dagger \cdot (1 - (\hat{\mu}_h e_h^\dagger)^\dagger))) \tag{5}$$

where $(\hat{\mu}_h e_h^\dagger)^\dagger$ denotes the error of the sub-SCN backpropagated to the hidden layer.

From Equation (5), it can be seen that the error at the input terminal of the ensemble classifier can be calculated as $E = \left(\sum_{h=1}^{n_h} ex_h\right)/n_h$. Therefore, the calculated $E$ would be backpropagated to the CNNs to optimize the model overall. To solve the problem of the changed dimension during feature reduction, and by learning from the upsampling method [23], zero is assigned to the positions, which are reduced in the process of $C^{l_w}$ to $B^{l_w,\beta_w}$; that is to say, the dimension of $B^{l_w,\beta_w}$ is expanded from $\delta$ to $K$ so as to calculate the backpropagation residuals as the reconstructed $\hat{B}^{l_w,\beta_w} = \left[\hat{B}_1^{l_w,\beta_w}, \ldots, \hat{B}_K^{l_w,\beta_w}\right]$.

## 4. The Intelligent Mechanism with Feedback Cognition for Greengage Grading Based on the Entropic Constraint of the Cognitive Error

The traditional machine cognition is generally open loop; that is, with the above trained feature extraction and ensemble cognition model, the corresponding cognition result of the testing database can be obtained. However, due to the randomness in the process of image sampling, feature extraction, and model training, the cognition and decision-making for greengage grading are uncertain. Therefore, the traditional open-loop method lacks evaluation of the cognitive result.

### 4.1. The Error Representation for Greengage Grading Based on the Entropy

The cognitive results for greengage grading with a determined feature level and feature efficiency should be measured according to the obtained greengage grades of the testing samples and the relative information of the training samples with the corresponding labels. The semantic information system of the cognitive error for greengage grading is constructed based on [24]. Then, the probability knowledge model for cognitive result evaluation can be established.

By assuming the current $w$th feedback cognition process with $l_w$ and $\beta_w$, for the input testing dataset of greengage image $Y = \{Y_1, \ldots, Y_d\}$, its corresponding grade label $O^{l_w,\beta_w} = \left\{O_1^{l_w,\beta_w}, \ldots, O_d^{l_w,\beta_w}\right\}$ would be obtained, where $O_t^{l_w,\beta_w}$ is the $t$th grade label of $Y_t$ and $t \in [1, d]$.

To measure the performance of the cognitive result under the current conditions, the semantic information system of the cognitive error for greengage grading for the testing sample $Y_t$ is constructed as $\Pi_t(w, l_w, \beta_w) = \left(U_{t,g}, M_t^g(w, l_w, \beta_w)\right)$, where $U_{t,g}$ is the $n_g$ dimensional semantic domain of the cognitive error for greengage grading between $Y_t$ and the corresponding training samples with the same class (the $g$th category) as the grade label $O_t^{l_w,\beta_w}$, and $M_t^g(w, l_w, \beta_w)$ is the $z_w$ dimensional semantic matrix of the cognitive error for greengage grading for $Y_t$ in the current process $g \in [1, o]$.

Regarding $\Pi_t(w, l_w, \beta_w)$, readers may refer to our previous work [25] for details. Here, $M_t^g(w, l_w, \beta_w)$ is an $n_g \times z_w$ matrix, which represents the performance of the cognitive result for $Y_t$ with the current model in the semantic space. Based on the equivalence relationship,

the quotient set $U_{t,g}/M_t^g(w, l_w, \beta_w) = \{E_{1,w}, \ldots, E_{s,w}\}$ can be gained by dividing $U_{t,g}$ by $M_t^g(w, l_w, \beta_w)$, where $E_{\sigma,w}$ is the $\sigma$th equivalence class, $\sigma \in [1, s]$, and $s \leq n_g$. It denotes a larger error in the current cognitive process for $Y_t$, with more elements in $U_{t,g}/M_t^g(w, l_w, \beta_w)$ and vice versa. Therefore, from the information theory, the cognitive process can be represented with an entropy function. The probability density function of the cognitive error distribution in the domain for the current process can be defined as in Equation (6):

$$P(E_{\sigma,w}) = \frac{|E_{\sigma,w}|}{|U_{t,g}|} \tag{6}$$

The corresponding entropy of the above probability distribution, namely the cognitive error entropy for greengage grading of the sample $Y_t$ during the $w$th feedback cognitive process with $l_w$ and $\beta_w$, is calculated as

$$H_t(w, l_w, \beta_w) = -\frac{\sum_{\sigma=1}^s P(E_{\sigma,w}) \log_2 P(E_{\sigma,w})}{\log_2 n_g} \tag{7}$$

In Equation (7), a smaller $H_t(w, l_w, \beta_w)$ indicates that the smaller uncertainty of $\Pi_t(w, l_w, \beta_w)$. Therefore, the cognition error of the testing sample $Y_t$ in the current cognition process is smaller and vice versa. Accordingly, the arbitrary regulation of $l_w$ and $\beta_w$ to make $H_t(w, l_w, \beta_w)$ extremely small in the feasible region is our goal for optimal cognition.

*4.2. The Feedback Regulation Mechanism of the Feature Efficiency and Feature Level Based on the Cognitive Error*

The calculation of the semantic error entropy for greengage grading quantitatively represents the system performance of the cognitive result with the determined feature level and feature efficiency. Thus, in this paper, $H_t(w, l_w, \beta_w)$ is calculated to correlate the training and testing processes. Furthermore, $l_w$ and $\beta_w$ are used as the regulative indexes of the model. The intelligent cognition for greengage grading with imitating the repeated comparison and inference would be realized though increasing the information level and controlling the classification quality.

In fact, the final cognitive result of a testing sample can be output with the cognitive performance that meets the target; otherwise, the feedback cognition is needed. Therefore, for the input testing dataset $Y$ in the $w$th feedback cognition process with $l_w$ and $\beta_w$, the sub-sample set $Y_w = \{Y_1, \ldots, Y_\psi\}$ is constructed first with the process which meets the target and outputs the cognitive result, where $Y_\alpha$ is the $\alpha$th sample obtaining the grade label in $Y_w$ and $\alpha \in [1, \psi]$. In this way, the samples with the process which could not satisfy the target currently and the corresponding calculation information can be extracted to reconstruct the testing dataset $Y$ according to the instruction $Y \leftarrow Y - Y_w$. Then, the optimal regulation of the feature level and feature efficiency can be achieved based on the reconstructed testing dataset. At this moment, the domain of the testing dataset involving parameter regulation is updated to $d \leftarrow d - \psi$.

Specific to the regulation of the feature efficiency within the level, $\beta_w$ with non-uniform regulation is more in line with the human cognition mechanism. In this paper, the incremental calculation model of classified accuracy is constructed based on $H_t(w, l_w, \beta_w)$, and the feature efficiency is adaptively regulated with the entropic constraint to enhance the fault tolerance and generalization of the model with a finite domain.

Suppose there is a current $w$th the feedback cognitive process with $l_w$ and $\beta_w$, and the calculated $H_t(w, l_w, \beta_w)$ of $Y_t$ when the feature level $l = l_w = l_{w+1}$; that is, the model level of the CNN remains unchanged. Then, the increment of the classified accuracy in the $w + 1$th feedback cognitive process is defined as in Equation (8):

$$\Delta\beta_{w+1} = \frac{l_{w+1} \cdot \min_{t=1,\ldots,d} [H_t(w, l_w, \beta_w)]}{w} \tag{8}$$

In this way, the regulation of the feature efficiency of the system in the $w + 1$th feedback cognitive process should abide by Equation (9):

$$\begin{cases} \beta_{w+1} = \beta_w + \cdot \beta_{w+1} \\ \beta_1 = 0.5, \beta_w \in (0.5, 1] \end{cases} \tag{9}$$

In this way, $\Delta\beta_w$ decreases with the increased $w$, which is closer to the sensory characteristics of human cognition from macro to micro.

Specific to the regulation of the feature level, as can be seen from the CNN calculation process, more hierarchical knowledge of greengage image can be extracted with the larger $l_w$, while the computation will also increase greatly. Therefore, it is generally desirable to obtain a relatively comprehensive feature space with the smallest feature level.

Suppose there is a current $w$th feedback cognition process with $l_w$ and $\beta_w$. If the cognitive error of the current system cannot meet the constraints of the entropy index along with repeated regulation of $\beta_w$, the optimal model parameters in the current level are reserved, and the regulation of the feature level in the $w + 1$th feedback cognition process should abide by Equation (10):

$$\begin{cases} l_{w+1} = l_w + 1 \\ l_1 = 1 \end{cases} \tag{10}$$

Thus, the deeper level of sample knowledge is mined to re-represent the greengage image.

## 5. The Feedback Cognitive Algorithm for Greengage Grading Based on Dynamic Feature and Ensemble Networks

Based on the above analysis, the feedback cognitive algorithm for greengage grading based on dynamic feature and ensemble networks is proposed in this paper to realize the feedback cognition of the greengage grade while imitating the human cognition mechanism. The pseudo-code is shown in Algorithm 1.

---

**Algorithm 1:** The feedback cognitive algorithm for greengage grading based on dynamic feature and ensemble networks

---

1.  Input: $U = \{U_1, \ldots, U_n\}$ and $Y = \{Y_1, \ldots, Y_d\}$. Set $Y_w = \{y_1, \ldots, y_\psi\}$ as all labeled testing samples in the $w$th cognition process, $\varepsilon$ as the expected error tolerance, $w_{max}$ as the maximum number of feedback, $m_{max}$ as the maximum number of the feature map, $l_{max}$ as the maximum number of the feature layer, and $\Delta\beta_{min}$ as the minimum number of $\Delta\beta_w$.
2.  Ensure: The cognitive result $R_{opt}$ and the optimal model parameter $P_{opt}$.
3.  $w \leftarrow 1$, $l_w \leftarrow 1$, $\beta_w \leftarrow 0.5$;
4.  **While** $Y \neq \phi$, **Then**
5.      **While** $w \leq w_{\max}$, and $l_w \leq l_{\max}$
6.          Obtain the optimal ensemble CNN-SCN model with $l_w$ and $\beta_w$ according to Equations (3)–(5);
7.          Obtain $O^{l_w, \beta_w}$
8.          **For** $t$ from 1 to $d$
9.            Calculate $H_t(w, l_w, \beta_w)$ according to Equation (7);
10.           **If** $H_t(w, l_w, \beta_w) \leq \varepsilon$, **Then**
11.             $R_t \leftarrow O_t^{l_w, \beta_w}$, $y_\alpha \leftarrow Y_t$;
12.           **End If**;
13.         **End For**;
14.     **End While**;
15.     Obtain $R_w$ and $Y_w$,
16.     Reconstruct $Y \leftarrow Y - Y_w$, and $d \leftarrow d - \psi$;
17.     **If** $\beta_w \leq 1$ and $\Delta\beta_w > \Delta\beta_{\min}$, **Then**
18.         Calculate $\Delta\beta_{w+1}$ and $\beta_{w+1}$ by Equations (8) and (9), and **continue**;
19.     **Else**, **Then**
20.         Calculate $l_{w+1}$ according to Equation (10), and **continue**;
21.     **End If**;
22. **End While**;
23. **return** Final $R_{opt}$ and $P_{opt}$ of testing dataset

---

## 6. Results and Discussion

To validate the performance of the proposed method, 3000 greengage images with a size of $56 \times 56$ were selected to build a sample database, including 5 grades with superior products, good products with scars, defective products, defective products with scars, and rotten products, which were used to carry out the simulation experiments. Then, 2500 greengage images were randomly selected as the training dataset $U$, and the remaining 500 greengage images were taken as the testing dataset $Y$ for 500 simulation experiments. Some samples of greengage grades are shown in Figure 4. The actual grades of the greengage samples are labeled by multiple sorters. All sampling experiments were run in MATLAB R2014b on a computer with a 2.90-GHz Intel Core i5-3380M processor and 4G memory. After many experimental attempts and a literature review, the expected error tolerance $\varepsilon = 0.01$, the maximum times for feedback cognition $w_{max} = 20$, the maximum number of convolutional feature maps $m_{max} = 20$, the maximum depth of the feature level $l_{max} = 5$, the minimum increment of the classified accuracy $\Delta\beta_{min} = 0.001$, the network parameters of the bagging SCN $\lambda = [1, 5, 10, 15, 20, 25, 30, 35, 40, 45, 50]$, the subnetwork number of SCNs $n_h = [2, 11]$ with a search step size of 1, and the number of SCN hidden layer nodes $L = [5, 50]$ with a search step size of 5 were selected as the preset parameters for the simulation experiments to find the most favorable performance.

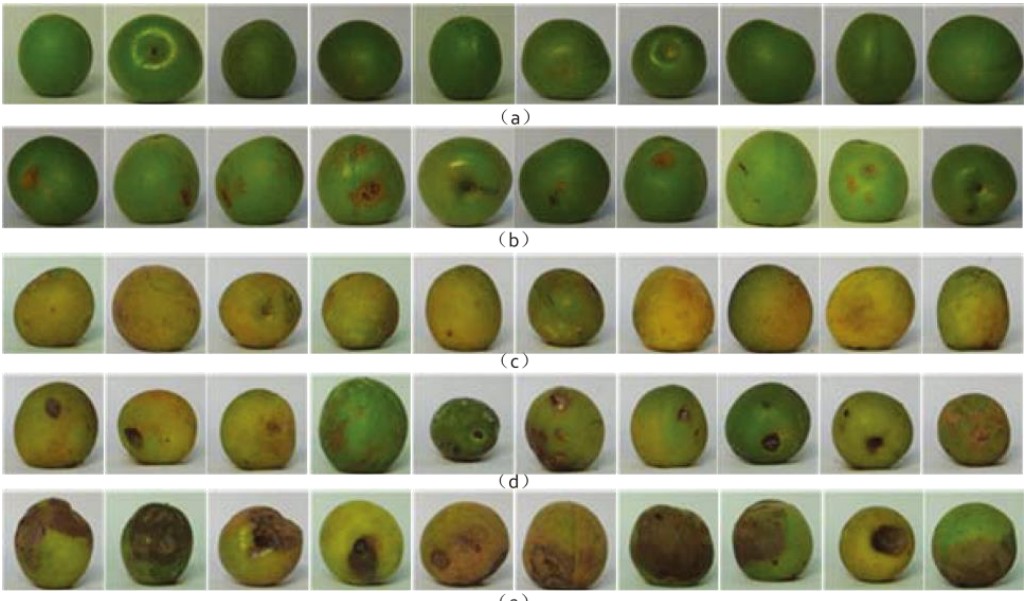

**Figure 4.** Some typical greengage samples: (**a**) superior products, (**b**) good products with scars, (**c**) defective products, (**d**) defective products with scars, and (**e**) rotten products.

Figure 5 shows the cognitive accuracy of the greengage grade $\Gamma$ for different parameters of the bagging SCN classifier, where $l = 2$, $m_1 = 7$, $m_2 = 5$, and $\beta = 0.76$. As can be seen in Figure 5a, the bagging SCN classifier had the global approximation capability of nonlinear mapping, and each hidden node in the network matched the different image features. Thus, $\Gamma$ would enhance with an increase in $L$ when $n_h = 1$. However, the cognitive performance may have been deteriorated with overfitting caused by the excess basis functions in the base network. That aside, each point in Figure 5b was obtained when the best $L$ value was used. As can be seen in Figure 5b, the adoption of the ensemble method could effectively enhance the reliability of the classification model to obtain a cognitive result with more robustness. However, not all $n_h$ values were appropriate. Having excessive basis functions not only did not improve the recognition accuracy, but it also incurred an additional computational cost. In our study, 6 SCN networks with 40 basis functions were sufficient to construct a reliable ensemble model.

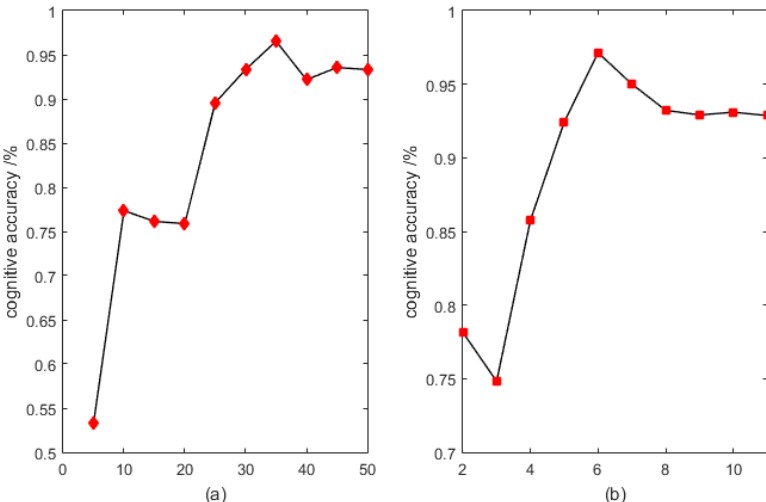

**Figure 5.** Cognitive accuracy vs. parameters of bagging SCN classifier: (**a**) performance with various $L$ values and (**b**) performance with various $n_h$ values.

Figure 6 shows the average cognition accuracy ρ for different training dataset domains $|U|$ with convolution feature extraction using our information entropy-based method and other fixed feature map-based methods ($m = 3$, $m = 6$, $m = 12$, and $m = 18$) in the open-loop experiment, in which the feature level $l = 1$ and fixed correspondence parameters were taken as an example. The information quantity required for classification increased with the sample domain, but the feature extraction method based on fixed feature maps obviously did not take this into account. Therefore, as indicated by the broken lines in Figure 6, the system performance was unitarily poor with $m = 3$ due to the small quantity of extracted feature information. When $m$ was 6, 12, and 18, relatively sufficient image information was extracted in a certain domain to improve the cognitive accuracy to a certain extent. However, for the cognitive problem within a limited domain, excessive extraction of feature information would lead to a deterioration in system performance, resulting in overfitting. As shown in Figure 6, when $m = 6$, $m = 12$, and $m = 18$, a better adaptation effect could be obtained around $|U| = 800$, $|U| = 1200$, and $|U| = 2000$, respectively. As shown by the solid line in Figure 6, our proposed method always built the feature space based on the maximum information quantity, so it had better performance for the classification with different domain sizes, especially in the large sample domain.

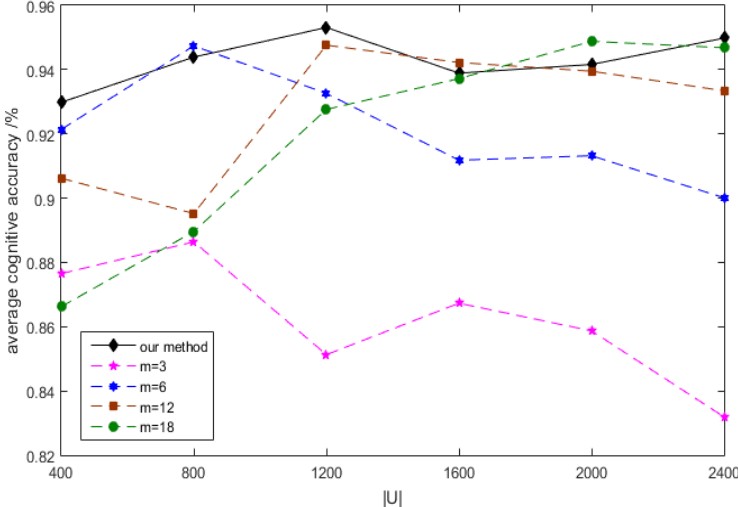

**Figure 6.** Average cognition accuracy vs. different training dataset domain $|U|$.

Figure 7 shows the cognitive accuracy of the testing dataset for different feedback cognition numbers $w$ with other fixed parameters in a closed-loop experiment. The coordinates of $w = 1$, $w = 2 \sim 4$, $w = 5 \sim 10$, $w = 11 \sim 16$, and $w = 17 \sim 20$ in the figure correspond to the system performance with the feature level $l = 1 \sim 5$, respectively. The cognitive results under the $l$ level were obtained based on the optimal parameters with the $l - 1$ level. From Figure 7, in the determined feature level, the feature appropriateness for the samples with a finite domain was controlled by regulating $\beta$ to improve the cognitive performance. The overall performance of each feature level correspondingly improved with the increasing $l$ value, which was due to the extraction of multi-level cognition knowledge from global to local through increasing the feature level during CNN modeling. However, when $l$ was close to $l_{\max}$, limited by the image size, and influenced by the image quality, a large amount of invalid and faulty information was added into the feature space, resulting in misclassification, and the computational complexity increased sharply with an increase in $l$. As a result, the system performance would be affected by deterioration since $l \geq 4$.

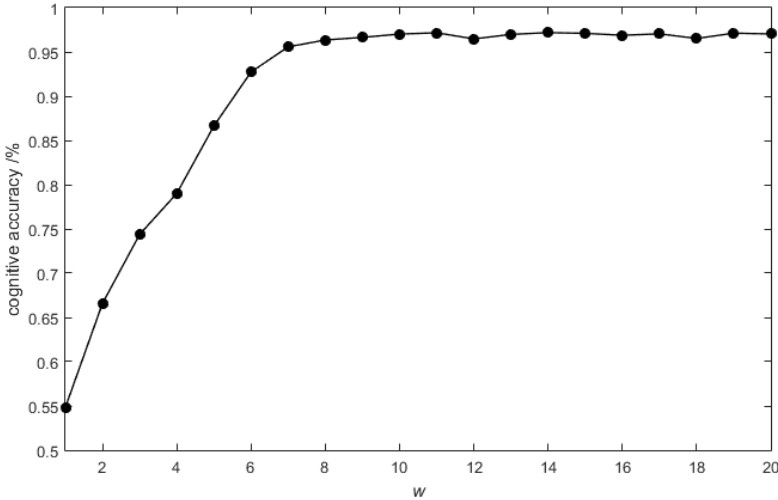

**Figure 7.** Cognitive accuracy vs. feedback cognition number $w$.

To prove the effectiveness of our method, the performance of several cognition methods were compared, including dynamic CNNs + Ensemble-RVFL [8], dynamic CNNs + SCN [9], dynamic CNNs + RVFL, a traditional CNN, color feature + quadratic discriminant analysis (QDA) [26], and Gabor wavelet + color moments + support vector machine (SVM) [27]. The average cognition accuracy and the average test time are listed in Table 1. All the comparative results are expressed as the mean $\pm$ standard deviation.

**Table 1.** Comparison of various methods.

| Methods | Average Recognition Accuracy (%) | Average Test Time (s) |
|---|---|---|
| Dynamic CNNs + bagging SCNs (our method) | $97.62 \pm 0.6$ | $10.47 \pm 1.9$ |
| Dynamic CNNs + SCN | $97.41 \pm 0.9$ | $10.29 \pm 1.3$ |
| Dynamic CNNs + Ensemble-RVFL | $97.48 \pm 1.2$ | $10.36 \pm 1.5$ |
| Dynamic CNNs + RVFL | $96.15 \pm 1.5$ | $9.67 \pm 1.2$ |
| Traditional CNN + softmax | $94.26 \pm 0.8$ | $5.53 \pm 1.5$ |
| Color feature + QDA | $93.91 \pm 0.9$ | $4.23 \pm 1.1$ |
| Gabor wavelet + color moments + SVM | $92.72 \pm 1.1$ | $4.41 \pm 0.9$ |

From Table 1, the following observations can be made.

The proposed greengage grading algorithm was effective and feasible, with an average cognition accuracy of 97.62%. The feedback cognition algorithm imitated the information interaction process of human cognition with repeated comparison and inference. Based on the entropy index, the credibility of the cognitive process and results for greengage grading were measured and constrained to regulate the feature level and feature efficiency, which realized the self-optimizing construction of the cognitive feature space and classified criteria for greengage grading in the sense of pattern classification. Therefore, the performance was better than that of the traditional open-loop cognition mode.

In addition, the algorithm complexity could be evaluated by the running time. The cognitive results of the greengage images with obvious features which met the index requirements could be obtained at a lower feature level and feature efficiency. However, for those similar samples near the classified surface, the cognitive results were gained by repeated cognition from global to local with the cognitive error calculation to dynamically optimize the cognitive criterion in the sense of pattern classification. In fact, this caused a lack of real-time performance under the premise of improving the cognitive accuracy. However, the overall system performance was better with the condition of a finite domain.

## 7. Conclusions

An intelligent cognition model for greengage grading based on dynamic feature and ensemble networks was explored in this paper. Cognitive knowledge for greengage grading is dynamically modeled with multi-level and multi-perspective approaches based on the adaptive feature level and maximum entropy. The entropy-based cognitive error calculation and the corresponding feedback mechanism for the feature level and feature efficiency regulation imitated the human cognition mechanism with repeated comparison and inference effectively. The experimental results show that our closed-loop dynamic feature modeling and ensemble network architecture can effectively improve the cognitive accuracy. In this paper, the classified accuracy parameter increased from 0.5 to 1 with variable steps at each feature level to improve the model's fault tolerance. However, during the experiment at a lower feature level, the regulated quantity tended to exceed the range of the parameter. Therefore, how to find the optimal classified accuracy in the determined feature level with the condition of a finite domain to optimize the system performance from the perspective of pattern classification is the focus of future research.

**Author Contributions:** Methodology and writing—original draft, K.C.; project administration, W.L.; validation, J.A.; writing—review and editing, T.B. All authors have read and agreed to the published version of the manuscript.

**Funding:** This research was funded by the Anhui Provincial Natural Science Foundation (grant number 1908085QF270) and the Talent Research Foundation of Hefei University (grand number 18-19RC43).

**Conflicts of Interest:** The authors declare no conflict of interest.

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
