# Peer review of "Greengage Grading Method Based on Dynamic Feature and Ensemble Networks"

_electronics, doi:10.3390/electronics11121832_

Round 1
Reviewer 1 Report
The acognitive method of greengage grade based on dynamic feature and ensemble networks is explored in this paper. The paper presents a relevant contribution.
However, some points need to be improved.
- The contributions of the paper need to be add in bullets in introduction section.
-algorithm 1 need to be better explianed
-authors need to better explain the parameters used in the experiments, values, range and preliminar tests.
Overall, the paper need to improve the english language.
Reviewer 2 Report
Overall, this work contains some publishable materials. I have two concerns with this submission, given as follows:
1. English writing needs further improvements. Some sentences are difficult to follow indeed. I suggest the authors to seek for professional native speakers for a proofreading to improve the readability.
2. This work has some overlaps to the reported result in [9]. The authors should stress the difference and highlight some new contributions from this work.
